# Learning Representation in Colour Conversion

## Abstract

Colours can be represented in an infinite set of spaces highlighting distinct features. Here, we investigated the impact of colour spaces on the encoding capacity of a visual system that is subject to information compression, specifically variational autoencoders (VAEs) where bottlenecks are imposed. To this end, we propose a novel unsupervised task: colour space conversion (*ColourConvNets*). We trained several instances of VAEs whose input and output are in different colour spaces, e.g. from RGB to CIE L*a*b* (in total five colour spaces were examined). This allowed us to systematically study the influence of input-output colour spaces on the encoding efficiency and learnt representation. Our evaluations demonstrate that ColourConvNets with decorrelated output colour spaces produce higher quality images, also evident in pixel-wise low-level metrics such as colour difference ($\Delta E$), peak signal-to-noise ratio (PSNR) and structural similarity index measure (SSIM). We also assessed the ColourConvNets' capacity to reconstruct the global content in two downstream tasks: image classification (ImageNet) and scene segmentation (COCO). Our results show 5-10% performance boost for decorrelating ColourConvNets with respect to the baseline network (whose input and output are RGB). Furthermore, we thoroughly analysed the finite embedding space of Vector Quantised VAEs with three different methods (single feature, hue shift and linear transformation). The interpretations reached with these techniques are in agreement suggesting that (i) luminance and chromatic information are encoded in separate embedding vectors, and (ii) the structure of the network's embedding space is determined by the output colour space.

## 1 Introduction

Colour is an inseparable component of our conscious visual perception and its objective utility spans over a large set of tasks such as object recognition and scene segmentation (Chirimuuta et al., 2015; Gegenfurtner & Rieger, 2000; Wichmann et al., 2002). Consequently, colour is an ubiquitous feature in many applications: colour transfer (Reinhard et al., 2001), colour constancy (Chakrabarti, 2015), style transfer (Luan et al., 2017), computer graphics (Bratkova et al., 2009), image denoising (Dabov et al., 2007), quality assessment (Preiss et al., 2014), to name a few. Progress in these lines requires a better understanding of colour representation and its neural encoding in deep networks. To this end, we present a novel unsupervised task: colour conversion.

In our proposed framework the input-output colour space is imposed on deep autoencoders (referred to as *ColourConvNets*) that learn to efficiently compress the visual information (Kramer, 1991) while transforming the input to output. Essentially, the output $y$ for input image $x$ is generated on the fly by a transformation $y = T(x)$, where $T$ maps input to output colour space. This task offers a fair comparison of different colour spaces within a system that learns to minimise a loss function in the context of information bottleneck principle (Tishby & Zaslavsky, 2015). The quality of output images demonstrates whether the representation of input-output colour spaces impacts networks' encoding power. Furthermore, the structure of internal representation provides insights on how colour transformation is performed within a neural network.

In this work, we focused on Vector Quantised Variational Autoencoder (VQ-VAE) (van den Oord et al., 2017) due to the discrete nature of its latent space that facilitates the analysis and interpretability of the learnt features. We thoroughly studied five commonly used colour spaces by training

ColourConvNets for all combinations of input-output spaces. First, we show that ColourConvNets with a decorrelated output colour space (e.g. CIE L*a*b) convey information more efficiently in their compressing bottleneck, in line with the presence of colour opponency in the human visual system. This is evident qualitatively (Figures 1 and A.1) and quantitatively (evaluated with three low-level and two high-level metrics). Next, we present the interpretation of ColourConvNets' latent space by means of three methods reaching a consensus interpretation: (i) the colour representation in the VQ-VAEs' latent space is determined by the output colour space, suggesting the transformation $T$ occurs at the encoder, (ii) each embedding vector in VQ-VAEs encodes a specific part of the colour space, e.g. the luminance or chromatic information, which can be modelled by a parsimonious linear transformation.

| Original | rgb2rgb | rgb2dkl | rgb2lab |
|---|---|---|---|

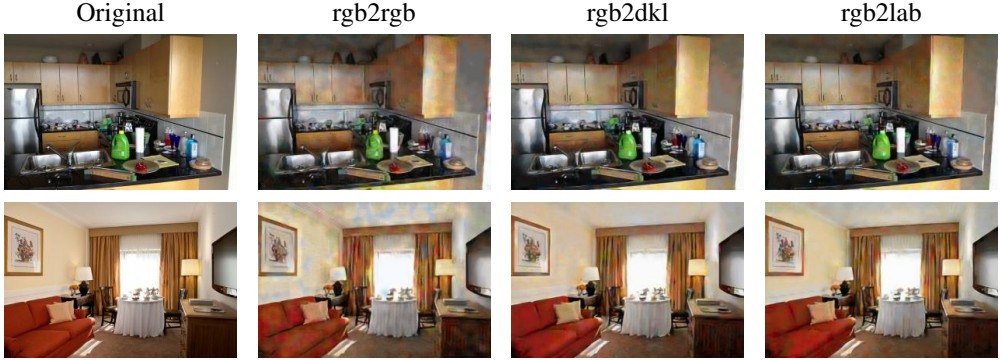

Figure 1: Qualitative comparison of three ColourConvNets (VQ-VAE of **K=8** and **D=128**). The first column is the networks' input and the other columns their corresponding outputs. The output images of *rgb2dkl* and *rgb2lab* have been converted to the RGB colour space for visualisation purposes. The artefacts in *rgb2rgb* are clearly more visible in comparison to the other ColourConvNets.

## 1.1 RELATED WORK

The effectiveness of different colour spaces have been investigated in a few empirical studies of deep neural networks (DNNs). Information fusion over several colour spaces improved retinal medical imaging (Fu et al., 2019). A similar strategy enhanced the robustness of face (Li et al., 2014; Larbi et al., 2018) and traffic light recognition (Cireşan et al., 2012; Kim et al., 2018). This was also effective in predicting eye fixation (Shen et al., 2015). Opponent colour spaces have been explored for applications such as style transfer (Luan et al., 2017; Gatys et al., 2017) and picture colourisation (Cheng et al., 2015; Larsson et al., 2016). Most of these works are within the domain of supervised learning. The most similar approach to our proposed ColourConvNets is image colourisation as a pretext task for unsupervised visual feature learning (Larsson et al., 2017).

Initial works on colour representation in DNNs revealed object classification networks learn to decorrelate their input images (Rafegas & Vanrell, 2018; Flachot & Gegenfurtner, 2018; Harris et al., 2019). This is a reminiscence of horizontal and ganglion cells that decorrelate retinal signal into colour-opponency before transmitting it to the visual cortex (Schiller & Malpeli, 1977; Derrington et al., 1984; Gegenfurtner & Kiper, 2003). Another set of works reported existence of hue-sensitive units (Engilberge et al., 2017) that mainly emerge in early layers (Bau et al., 2017). Representation of colours in deep networks at intermediate and higher layers is rather understudied. In this article, we specifically focus on the intermediate representation that emerges at the latent space of autoencoders, which to the best of our knowledge has not been reported in the literature.

## 2 COLOUR CONVERSION AUTOENCODERS

In this article, we propose a novel unsupervised task of colour conversion: the network's output colour space is independent of its input (see Figure 2). A colour space is an arbitrary definition of colours' organisation in the space (Koenderink & van Doorn, 2003). Thus, the choice of transfor-

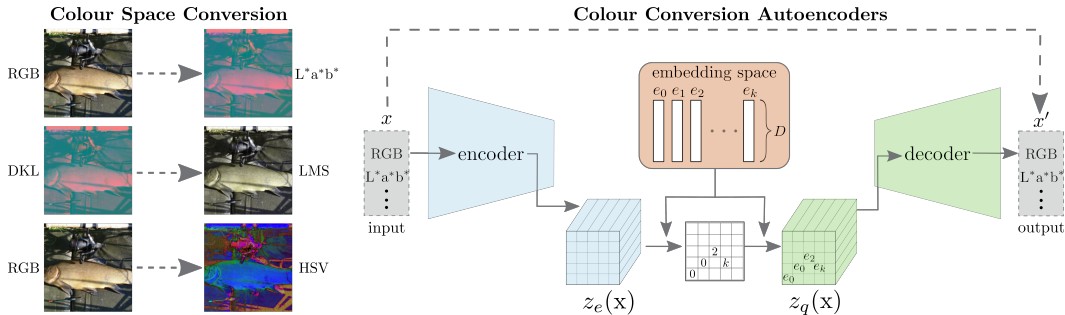

Figure 2: Left: exemplary conversions across different colour spaces. Right: the schematic view of VQ-VAE ColourConvNets.

mation matrix $T$ in ColourConvNets is perfectly flexible to model any desired space,

$$\mathcal{C}_{in} \xrightarrow{\ T\ } \mathcal{C}_{out}, \tag{1}$$

where $\mathcal{C}_{in}$ and $\mathcal{C}_{out}$ are the input and output colour spaces. This framework offers a controlled environment to compare colour spaces within a complex visual system. Here, we studied their effectiveness in information encoding constrained to a bottleneck. This can be extended to encompass other constraints (such as entropy, energy, wiring, etc.) relevant to understanding colour representation in complex visual systems. We further used this structure to compare autoencoder's latent space across colour spaces aiming to decipher the intermediate colour representation within these networks. The proposed framework can also be employed in applications, e.g., as an add-on optimisation capsule to any computer vision application (Mosleh et al., 2020), or as a proxy task for visual understanding (Larsson et al., 2017).

## 2.1 NETWORKS

We studied a particular class of VAEs—Vector Quantised Variational Autoencoder (VQ-VAE) (van den Oord et al., 2017)—due to the discrete nature of its latent embedding space that facilitates the analysis and interpretability of the learnt features, which distinguishes it from others (Kingma & Welling, 2013). VQ-VAE consists of three main blocks: 1) an encoder that processes the input data $x$ to $z_e(x)$; 2) a latent embedding space $\{e\} \in \mathbb{R}^{K \times D}$, with $K$ vectors of dimensionality $D$, that maps $z_e(x)$ onto $z_q(x)$ by estimating the nearest vector $e_i$ to $z_e(x)$; 3) a decoder that reconstructs the final output $x'$ with a distribution $p(x|z_q(x))$ over the input data (see the right panel in Figure 2). The loss function is defined as follows,

$$L = \log p(x|z_q(x)) + \|sg[z_e(x)] - e\|_2^2 + \beta \|z_e(x) - sg[e]\|_2^2, \tag{2}$$

where $sg$ denotes the stop gradient computation that is defined as the identity during the forward-propagation, and with zero partial derivatives during the back-propagation to refrain its update. The first term in Eq. 2 corresponds to the reconstruction loss incorporating both encoder and decoder; the second term updates the embedding vectors; and the third term harmonies the encoder and embedding vectors. The parameter $\beta \in \mathbb{R}$ is set to $0.5$ in all our experiments.

## 2.2 COLOUR SPACES

We explored five colour spaces: RGB, LMS, CIE L*a*b*, DKL and HSV. The standard space in electronic imaging is RGB that represents colours by three additive primaries in a cubic shape. The LMS colour space corresponds to the response of human cones (long-, middle-, and short-wavelengths) (Gegenfurtner & Sharpe, 1999). The CIE L*a*b* colour space (luminance, red-green and yellow-blue axes) is designed to be perceptually uniform (CIE, 1978). The DKL colour space (Derrington-Krauskopf-Lennie) models the opponent responses of rhesus monkeys in the early visual system (Derrington et al., 1984). The HSV colour space (hue, saturation, value) is a cylindrical representation of RGB cube designed by computer graphics.

The input-output to our networks can be in any combination of these colour spaces. Effectively, our VQ-VAE models, in addition to learning efficient representation, must learn the transformation

function from their input to output colour space. It is worth considering that the original images in explored datasets are in the RGB format. Therefore, one might expect a slight positive bias towards this colour space given its gamut defines the limits of other colour spaces.

## 3 EXPERIMENTS

We trained several instances of VQ-VAEs with distinct sizes of embedding space $\{e\} \in \mathbb{R}^{K \times D}$. The training procedure was identical for all networks: trained with Adam optimiser (Kingma & Ba, 2014) ($lr = 2 \times 10^{-4}$) for 90 epochs. To isolate the influence of random variables, all networks were initialised with the same set of weights and an identical random seed was used throughout all experiments. We used ImageNet dataset (Deng et al., 2009) for training. This is a visual database of object recognition in real-world images, divided into one thousand categories. The training set contains 1.3 million images. At every epoch, we exposed the network to 100K images of size $224 \times 224$ of three colour channels. Figure B.1 reports the progress of loss function for various ColourConvNets. A similar pattern of convergence can be observed for all trained networks.

To increase the generalisation power of our findings, we evaluated all networks on the validation-set of three benchmark datasets: ImageNet (50K images), COCO (5K images), and CelebA (~20K images). COCO is a large-scale object detection and segmentation dataset (Lin et al., 2014). CelebA contain facial attributes of celebrities (Liu et al., 2015). We relied on two classes of evaluation[1]: low-level (Theis et al., 2015), capturing the local statistics of an image; high-level (Borji, 2019), assessing the global content of an image.

**Low-level evaluation** – We computed three commonly used metrics to measure the pixel-wise performance of networks: (i) the colour difference CIE $\Delta E$-2000 (Sharma et al., 2005), (ii) peak signal-to-noise ratio (PSNR), and (iii) structural similarity index measure (SSIM) (Wang et al., 2004).

**High-level evaluation** – Pixel-wise measures are unable to capture the global content of an image and whether semantic information remains perceptually intact. To account for this limitation, we performed a procedure similar to the standard Inception Score (Salimans et al., 2016; Borji, 2019) by feeding the reconstructed images to two pretrained networks (without fine-tuning) that perform the task of object classification, ResNet50 (He et al., 2016), and scene segmentation, Feature Pyramid Network—FPN (Kirillov et al., 2019). ResNet50 and FPN expect RGB inputs, thus non-RGB reconstructions were converted to RGB. The evaluation for ResNet50 is the classification accuracy on ImageNet dataset. The evaluation for FPN is the intersection over union (IoU) on COCO dataset.

### 3.1 EMBEDDING SIZE

We first evaluated the influence of embedding size for four regimes of ColourConvNets whose input colour space is the original RGB images. The low-level evaluation for ImageNet is reported in Figure 3 and COCO Figure C.1. Across three metrics, the poor performance of *rgb2hsv* pops up at low-dimensionality of the embedding vector ($D = 8$). This might be due to the circular nature of hue. For the smallest and the largest embedding space, we observe no significant differences between the four networks. However, for embedding spaces of $8 \times 8$ and $8 \times 128$ an advantage appears for networks whose outputs are opponent colour spaces (DKL and CIE L*a*b).

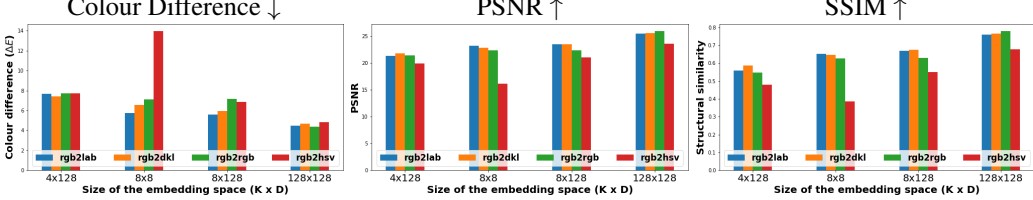

Figure 3: Low-level evaluation for embedding spaces of different size (ImageNet validation-set).

The corresponding high-level evaluation is reported in Figure 4. The overall trend is much alike for both tasks. The lowest performance occurs for *rgb2hsv* across all embedding spaces. Colour-

---

[1]For reproduction, the source code and all experimental data are available in the supplementary materials.

ConvNets with an opponent output colour space systematically perform better than *rgb2rgb*, with an exception for the largest embedding space ($128 \times 128$) where all networks perform equally (despite the substantial compression, 70% top-1 accuracy on ImageNet and 60% IoU on COCO). The comparison of low- and high-level evaluation for the smallest embedding space ($4 \times 128$) (Figure 4 versus Figures 3 and C.1) demonstrates the importance of high-level evaluation. Although no difference emerges for the low-level measure, the classification and segmentation metrics are substantially influenced by the quality of the reconstructed images in those four VQ-VAEs.

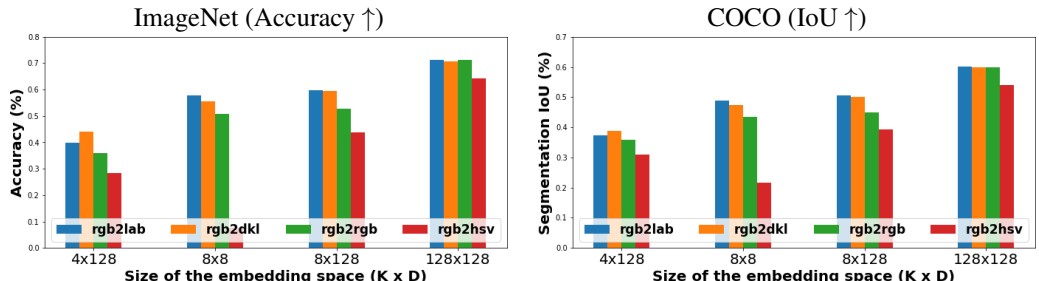

Figure 4: High-level visual task evaluation. Left: ResNet50's classification accuracy on reconstructed images of ImageNet. Right: FPNS's segmentation IoU on reconstructed images of COCO.

## 3.2 PAIRWISE COMPARISON

For the two embedding spaces with the largest differences ($8 \times 8$ and $8 \times 128$) we conducted an exhaustive pairwise comparison across two regimes of colour spaces: sensory (RGB and LMS) versus opponency (DKL and CIE L*a*b*). HSV is excluded in these analysis due to the aforementioned reason. Figure 5 presents the low-level evaluation results for ImageNet (COCO Figure C.2 and CelebA Figure C.3). There is a clear tendency of better performance for ColourConvNets with an opponent output colour space across all measures and datasets. Overall, the *rgb2lab* reconstructs the highest quality images. In comparison to the baseline (i.e. *rgb2rgb*) both *rgb2lab* and *rgb2dkl* obtain substantially lower colour differences, and higher PSNRs and SSIMs.

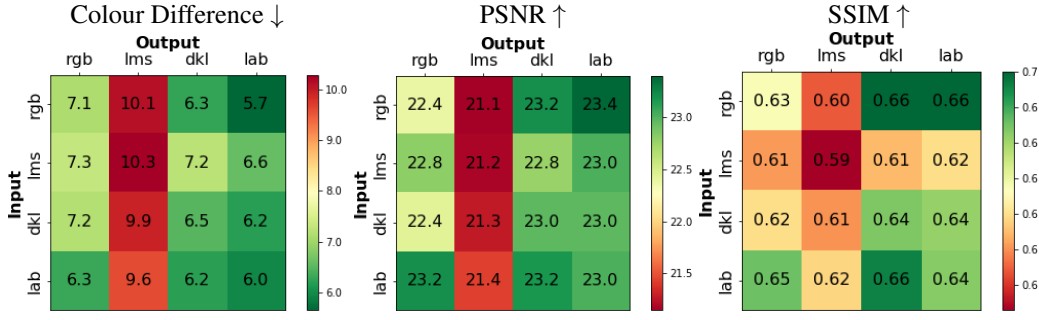

Figure 5: Low-level pairwise comparison of two groups of input-output colour spaces (ImageNet validation-set). Figures are averaged over two embedding spaces $8 \times 8$ and $8 \times 128$.

The high-level evaluation results are reported in Figure 6. In agreement to previous findings, *rgb2lab* performs best across both datasets and embedding spaces. Overall, ColourConvNets with an opponent output space show a clear advantage: *rgb2lab* and *rgb2dkl* obtain 5-7% higher accuracy and IoU with respect to the baseline *rgb2rgb*.

## 4 PERFORMANCE ADVANTAGE

The main difference between two regimes of colour spaces (sensory versus opponency) is their intra-axes correlation. The intra-axes correlation for RGB and LMS is very high, hence referred to as *correlated* colour spaces. On the contrary, the intra-axes correlations for CIE L*a*b* and DKL

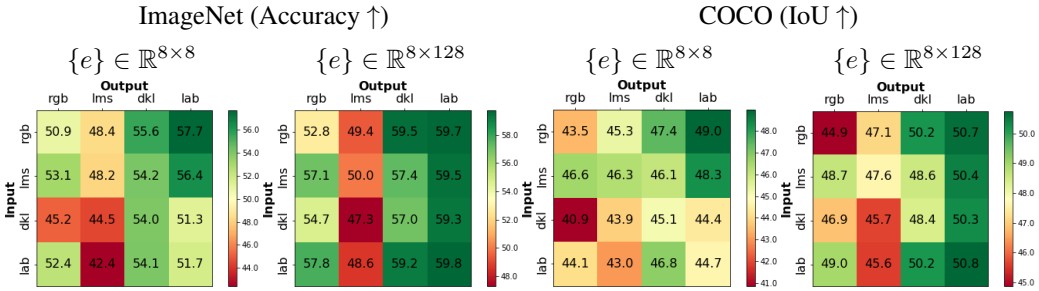

Figure 6: High-level pairwise comparison of sensory (RGB and LMS) versus opponency (DKL and CIE L*a*b) input-output colour spaces of VQ-VAEs.

are very low, hence referred to as *decorrelated* colour spaces.[2] In biological visual systems, the retinal signal is transformed to opponency before transmitted to the visual cortex through the LGN bottleneck (Zhaoping, 2006b). This transformation has been argued to boost the efficiency of information coding (Buchsbaum & Allan, 1983; Ruderman et al., 1998; Lee et al., 2001). Here, our results show a similar phenomenon in deep autoencoders that compress information in their bottleneck. Contrary to this, the ImageNet classification performance was reported unaltered when input images converted from RGB to CIE L*a*b* (Mishkin et al., 2017). This might be explained by the lack of bottleneck constraint in their examined architecture, thus decorrelating colour representation leads to no extra advantage. Interestingly, we can observe this with ColourConvNets of largest embedding space ($128 \times 128$), suggesting decorrelation of colour signal become beneficial when system is constrained in its information flow.

Previous works in the literature (Foster et al., 2008; Malo, 2019) have measured the decorrelation characteristics of colour opponent spaces in information theoretical analysis and demonstrated their effectiveness in encoding natural images. The understanding of how a complex visual system, driven by error minimisation strategy (Laparra et al., 2012), might utilise these properties at the system level is of great interest (Lillicrap & Kording, 2019). We hypothesised that an efficient system distributes its representation across all resources instead of heavily relying on a few components (Laughlin, 1981). To measure this, the histogram of embedding vectors across all images of ImageNet (50K) and COCO (5K) were computed. A zero standard deviation in the frequency of selected vectors means embedding vectors are equally used by the network. Figure 7 reports the error rate as a function of this measure. A significant correlation emerges in both datasets, suggesting a more uniform contribution of embedding vectors enhances visual encoding in VQ-VAEs. This matches the neural model of histogram equalisation (Pratt, 2007; Bertalmío, 2014) and is consistent with the efficient coding theory for the biological visual system (Barlow, 1961; Zhaoping, 2006a).

## 5 INTERPRETING THE EMBEDDING SPACE

Comprehension of the features learnt by a DNN remains a great challenge to the entire community (Lillicrap & Kording, 2019). Generative models and in particular variational autoencoders are no exceptions. Strategies on the interpretation of the latent space structure include interpolation in latent space arithmetic operations on learnt features (Radford et al., 2015; Bojanowski et al., 2017; Kim et al., 2018). In practice, however, these approaches require explicit human supervision, a cumbersome task due to the often large dimensionality of the latent space. Here, we borrowed the "lesion" technique, commonly practised in the neuroscience community (Vaidya et al., 2019), and applied it to the embedding space by silencing one vector at a time (i.e. setting its weights to zero). This procedure is referred to as "ablation" in the learning community and it has been useful in dissecting classification DNNs (Sandler et al., 2018) and GANs (Bau et al., 2020). To measure the consequences of vectors' lesion, we analysed the ColourConvNets' embedding space with three distinct methods: (i) single features, (ii) linear transformation and (iii) hue-shift.

---

[2]We computed these correlations $r$ in all images of ImageNet dataset (hundred-random pixels per image). RGB: $r^{RG} \approx 0.90$, $r^{RB} \approx 0.77$, $r^{GB} \approx 0.89$; LMS: $r^{LM} \approx 1.00$, $r^{LS} \approx 0.93$, $r^{MS} \approx 0.93$; L*a*b*: $r^{L*a*} \approx -0.14$, $r^{L*b*} \approx 0.13$, $r^{a*b*} \approx -0.34$; DKL: $r^{DK} \approx 0.01$, $r^{DL} \approx 0.14$, $r^{KL} \approx 0.61$.

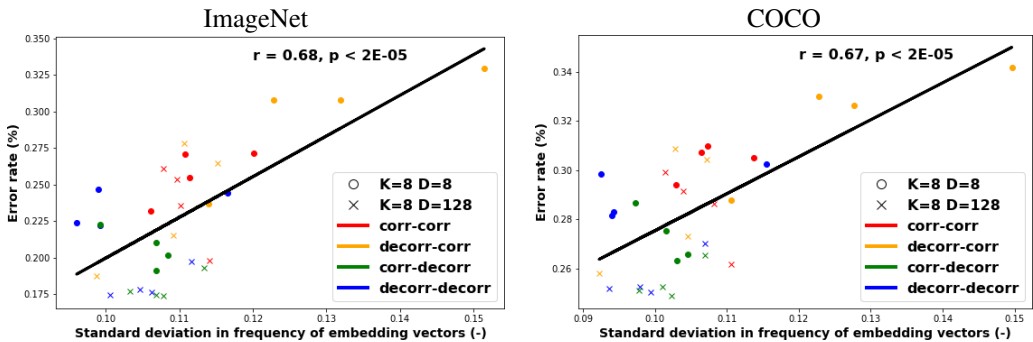

Figure 7: Error rate as a function of the difference in frequency of selected vectors in the embedding space. A value of zero in the $x$-axis indicates all embedding vectors are equally used by the model. Higher values of $x$ indicate that the model relies heavily on certain vectors.

## 5.1 SINGLE FEATURES

To visualise the encoded representation by each embedding vector, we sampled from the embedding space an example of spatial size $2 \times 2$ with all cells set to the same vector index. Figure 8 shows the reconstructed images for all network combinations with embedding space $\{e\} \in \mathbb{R}^{8 \times 128}$ (Figure D.1 for $\{e\} \in \mathbb{R}^{8 \times 8}$). The input colour space is the same in each row, and the output space is the same in each column. An interesting column-wise feature appears. Networks with an identical output colour space share a similar set of hues arranged in a different order. The order within the embedding space of VQ-VAEs is arbitrary and changing it does not impact the network's output. This is an interesting phenomenon suggesting: (i) the colour representation in network's embedding space is an attribute of its output colour space, and (ii) the colour transformation $T$ is performed by encoder before reaching the embedding space. This is an exciting line of investigation for feature studies to systematically explore whether the concept of unique hues and colour categories (Witzel & Gegenfurtner, 2018; Siuda-Krzywicka et al., 2019) emerge in machine colour representation.

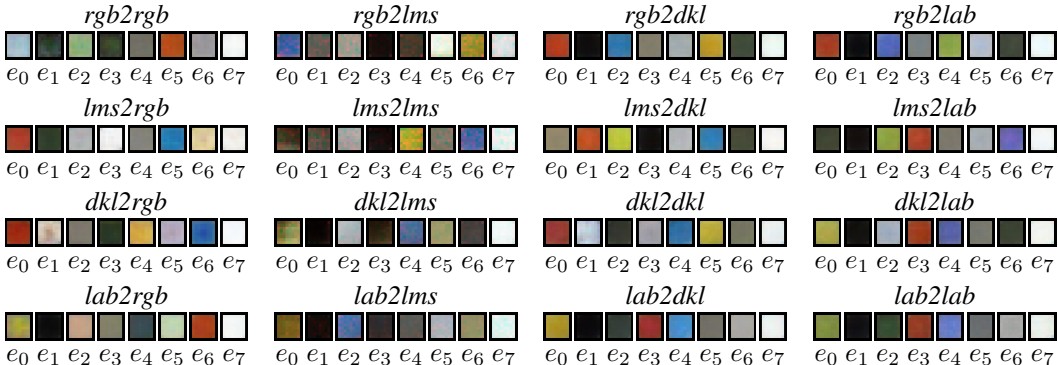

Figure 8: The reconstruction output by selecting a single vector of the entire embedding space. All models are VQ-VAE of **K=8** and **D=128**.

The samples reconstructed with a single embedding vector are not perfectly uniform (some small spatial variation is visible in Figure 8). To better understand the spatio-chromatic aspect of the encoded information, we again drew a sample of spatial size $2 \times 2$ from the embedding space; this time instead of setting all elements to a single vector, we combined two vectors in different spatial directions. The resulting reconstruction for the *rgb2lab* is illustrated in Figure 9. The embedding space spatial direction is relayed to the networks reconstructed images although the degree of it depends on the pair of embedding vectors. For instance, the horizontal combination of $e0 - e7$ results in two stripes of colour, while $e0 - e2$ turn into three stripes. This is naturally due to the fact that embedding vectors encode beyond chromatic information, but also the distinct nature of spatio-chromatic combination the decoder learns.

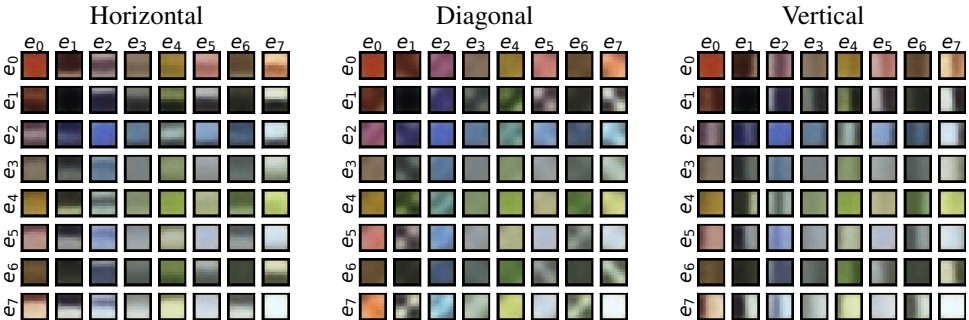

Figure 9: The reconstruction by a pairwise combination of embedding vectors in different spatial directions. ColourConvNet *rgb2lab* with **K=8** and **D=128**. In all cases a sample of spatial size $2 \times 2$ was drawn from the embedding space. Horizontal: the top elements set to vector $e_i$ and bottom $e_j$. Diagonal: the principal diagonal $e_i$ and off-diagonal $e_j$. Vertical: the left elements $e_i$ and right $e_j$.

## 5.2   LINEAR TRANSFORMATION

Three exemplary reconstructions by *rgb2dkl* network are illustrated in Figure 10 (for other Colour-ConvNets refer to Sec. D.2). Panel **A** corresponds to the full embedding space and **B–D** show examples of reconstructions with distinct vector lesions causing clear visible effects. In **B**, only the lightness of bright pixels is reduced (attend to pixels outside the window and around light bulbs). In **C & D**, lesioning $e_0$ and $e_2$, turns reddish and blueish pixels into achromatic. This is in agreement to colour of *rgb2dkl* $e_0$ and $e_2$ in Figure 8.

We hypothesised that the changes induced by a lesion could be approximated by a linear transformation mapping the pixel distribution of the full reconstruction onto the lesion image. To compute these transformations, we used a multi-linear regression finding the best linear fit for the 1% of most affected pixels. The resulting $3 \times 3$ matrix is a linear transformation in CIE L*a*b colour space. We have illustrated the result of applying these linear transformations on the right side of Figure 10. Panel **E** corresponds to the full RGB cube (essentially the CIE L*a*b* planes limited by RGB gamut). In **F–H** the very same points are plotted transformed by the model of lesioned vector.

Overall, lesions are closely approximated by a linear transformation: on average accounting for 97% of the total variance in the lesion effect (the lowest bound was 86%). This visualisation offers an intuitive interpretation of the learnt representation within the embedding space. In the images of the second row (panel **B**), contrast in bright pixels is reduced and colour is little modified. We can observe this in its corresponding CIE L*a*b* planes (e.g. attend the a*b* plane in **F** where the overall chromaticity structure is retained). In **C**, red pixels turn grey also evident in its corresponding CIE L*a*b* planes (panel **G**) where red coordinates are collapsed.

The geometrical properties of a transformation can be captured by the relative norms of its eigenvalues. For instance, zero-value eigenvalues indicate the extreme case of a singular matrix, corresponding to a linear transformation projecting a three-dimensional space onto lower dimensions. We quantified this by defining a singularity index (Philipona & O'regan, 2006). Consider a transformation matrix $T$ approximating the lesion effect on the image colour distribution. Let $\lambda_1$, $\lambda_2$ and $\lambda_3$ be the three eigenvalues of $T$, such that $\|\lambda_1\| > \|\lambda_2\| > \|\lambda_3\|$. The singularity index is defined as: $SI = 1 - \frac{\lambda_3}{\lambda_1}$. This index captures the essence of these transformations. On the one hand, the low value of $SI$ in **F** suggests the global shape of colour space is retained while its volume is reduced. On the other hand, high values of $SI$ in panels **G** and **H** indicate the near collapse of a dimension.

## 5.3   HUE SHIFT

We further quantified the impact of vector lesion by computing the difference in CIE L*a*b* between the full reconstructed image and lesioned one. The average difference over all pixels for *rgb2dkl* is illustrated in Figure 11 (refer to Sec. D.3 for other ColourConvNets). The results of hue shift analysis restate the interpretation of learnt representation. For instance, the direction of shift in $e_0$ is limited to the first quadrant of the chromaticity plane (red pixels). The $e_1$ vector largely

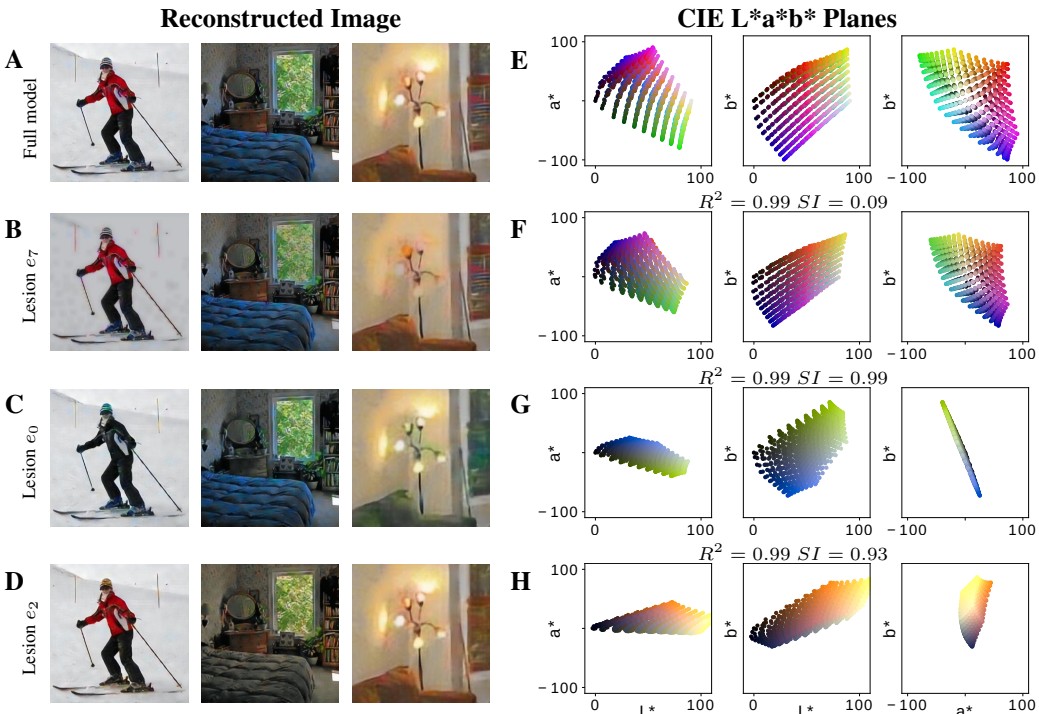

Figure 10: The lesion effect visualisation for the *rgb2dkl* $\{e\} \in \mathbb{R}^{8 \times 128}$. Left, reconstructed images by **A:** the full model; **B–D:** the lesion embedding space. Right, scatter plots of pixels in CIE L*a*b* coordinates of **E:** the entire RGB cube; **F–H**: after applying the linear model to the RGB cube.

encodes the low-luminance information (the negative direction in the L* axis). The $e_2$ vector predominantly influences the blue pixels (the negative direction in the b* axis). Similar colours emerge for *rgb2dkl* $e_0$, $e_1$ and $e_2$ in Figure 8.

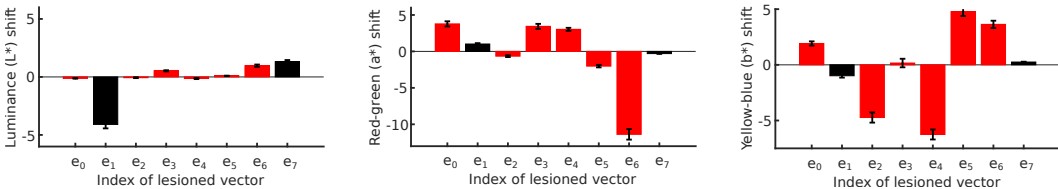

Figure 11: Average hue shifts for *rgb2dkl* $\{e\} \in \mathbb{R}^{8 \times 128}$ in CIE L*a*b* coordinates. Black- and red-bars indicate significant impact on the luminance- or chromatic-channels respectively.

## 6 CONCLUSION

We proposed the unsupervised *colour conversion* task to investigate colour representation in deep networks. We studied the impact of colour on the encoding capacity of autoencoders, specifically VQ-VAEs whose feature representation is constrained by a discrete bottleneck. The comparison of several ColourConvNets exhibits advantage for a decorrelated output colour space. This is evident qualitatively and measured quantitatively with five metrics. We discussed this benefit within the framework of efficient coding and histogram equalisation. These findings might contribute to our understanding of why the brain's natural network has developed the opponent representation. We further explored the networks' internal representation by means of three methods. Our analyses suggest: (i) the colour transformation is performed at the encoding stage prior to reaching the embedding space, (ii) despite the spatio-chromatic nature of the constituent vectors, many manifest a clear effect along one colour direction that can be modelled by a parsimonious linear model.

### ACKNOWLEDGEMENTS

Use unnumbered third level headings for the acknowledgments. All acknowledgments, including those to funding agencies, go at the end of the paper.

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

# A    QUALITATIVE COMPARISON

Along with the quantitative evaluations reported in the manuscript, the benefits of utilising a decorrelated colour space for the network's output can be appreciated qualitatively (see Figure A.1). These are representative samples from the COCO dataset Lin et al. (2014). The Jupyter-Notebook scripts in the supplementary materials provide more examples[3]. Overall, the *rgb2dkl* and *rgb2lab* VQ-VAEs generate more coherent images. For instance, in the first row of Figure A.1, the *rgb2rgb* output contains a large amount of artefacts on walls and ceilings. In contrast, the output of *rgb2dkl* and *rgb2lab* are sharper.

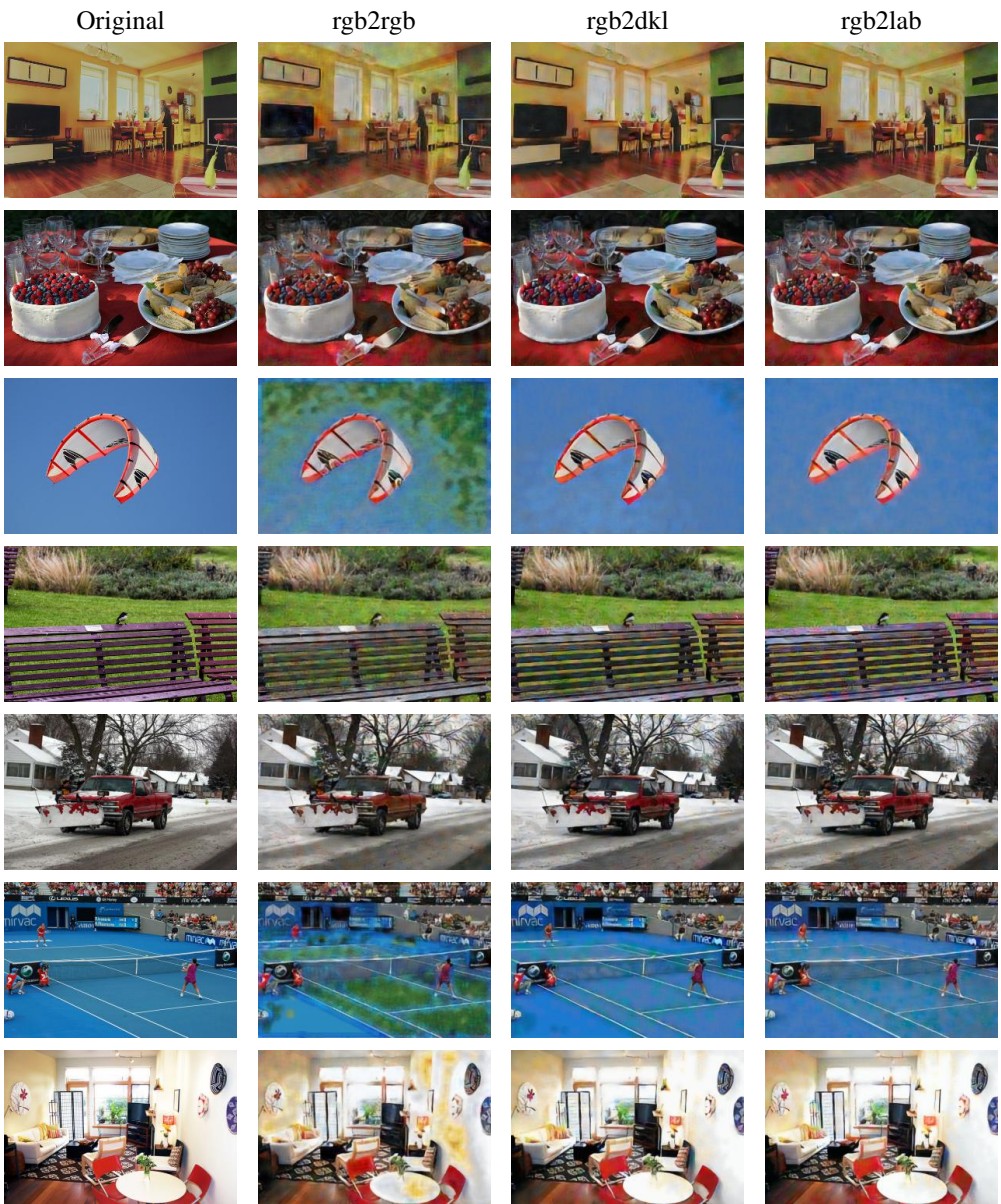

Figure A.1: Qualitative comparison of three VQ-VAEs of **K=8** and **D=128**.

---

[3]The weights of all trained networks and image outputs of lesion study exceed the 100MB upload limit, but they are publicly available for interested readers under this link `https://www.dropbox.com/sh/e1l3p3uot94q0fy/AADg0rmxyiC3UNifTtqIpg2Pa?dl=0`.

# B  LOSS FUNCTION

The loss function (Eq. 2) is computed in ColourConvNets' output colour space between the ground-truth and network's output. Figure B.1 reports the evolution of losses for all VQ-VAEs of **K=8** and **D=128**. The convergence of losses is comparable across all networks regardless of their input-output colour space.

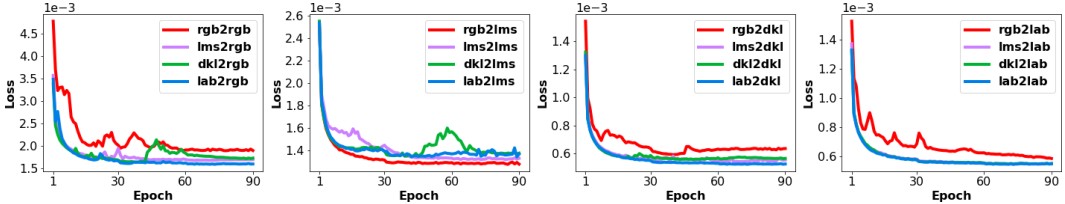

Figure B.1: Evolution of losses for VQ-VAEs of **K=8** and **D=128**. In each panel, the ColourConvnets have the same output space. Across panels, curves of the same colour have the same input space.

# C  LOW-LEVEL EVALUATION

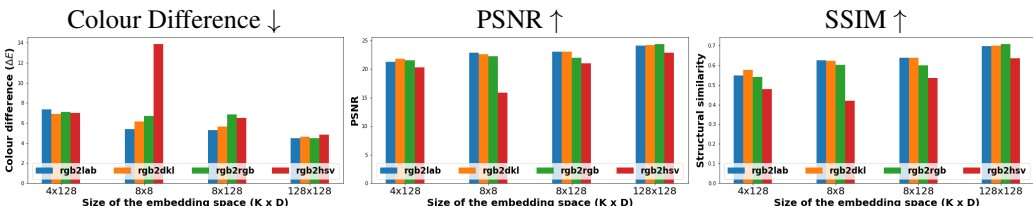

Figure C.1: Low-level evaluation for embedding spaces of different size (COCO validation-set).

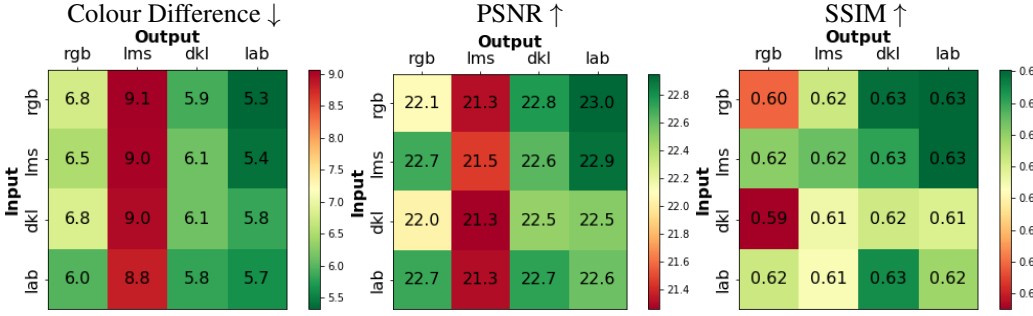

Figure C.2: Low-level pairwise comparison of two groups of input-output colour spaces (COCO validation-set). Figures are averaged over two embedding spaces $8 \times 8$ and $8 \times 128$.

# D  INTERPRETING THE EMBEDDING SPACE

## D.1  SINGLE FEATURES

The hues obtained from single vectors of VQ-VAE with **K=8** and **D=8** is reported in Figure D.1. The effect observed for larger ColourConvNets (i.e. networks with an identical output colour space sharing a similar set of hues arranged in a different order) is less evident here. This might be due to the dimensionality of the embedding space. This regime consists of vectors of 8 elements, whereas in the previous regime (Figure 8) the dimensionality of vectors is 128.

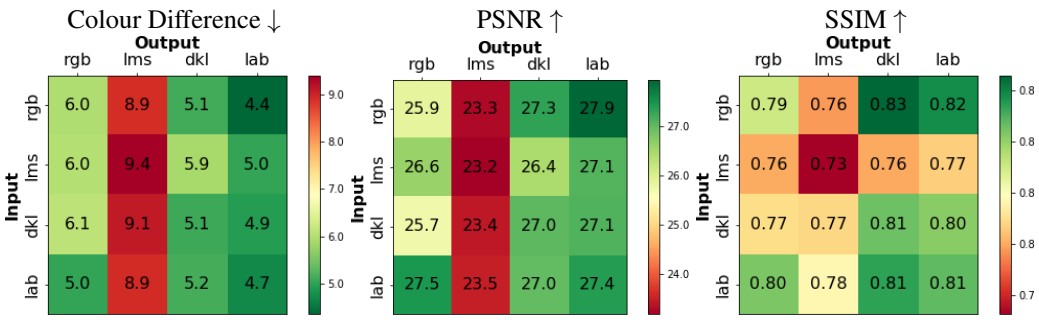

Figure C.3: Low-level pairwise comparison of two groups of input-output colour spaces (CelebA validation-set). Figures are averaged over two embedding spaces $8 \times 8$ and $8 \times 128$.

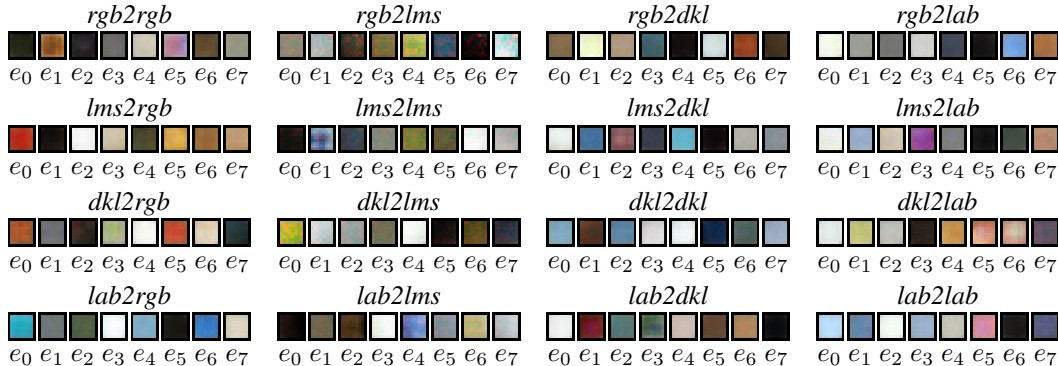

Figure D.1: The reconstruction output by selecting a single vector of the entire embedding space. All models are VQ-VAE of **K=8** and **D=8**.

## D.2 LINEAR MODELLING OF VECTOR LESIONS

In order to understand the features learnt by the colour conversion networks, we exercised the "lesion" technique. It consists of silencing the embedding space's vectors one at a time. We explored whether a vector lesion can be modelled by a simple linear transformation. We estimated the transformation matrix that maps the pixel distribution of the full reconstruction onto the lesion image (refer to Section 4 in the manuscript). To our surprise, this simple parsimonious modelling can capture a large portion of the vector's encoding. We present qualitative results for VQ-VAEs with $K = 8$ and $D = 128$ in the colour conversion networks *rgb2dkl* (Figure D.2), *rgb2lab* (Figure D.3) and *rgb2rgb* (Figure D.4).

The top row of Figure D.2 illustrated three examples from the COCO dataset reconstructed with the full model of *rgb2dkl*. The following rows depict the reconstruction output of lesion technique exercised on each embedding vector $e_i$ (the "Lesion output" column), alongside with the linear modelling estimation (the "Linear model" column) obtained by applying the linear transformation to the full reconstructed image. On the bottom right corner, we have reported the fitness of the mode, correlation in the CIE L*a*b* colour coordinates ($r$) between the colour pixels of the lesion output and the linear model. The overall fitness is very high for such a simple model. Even in cases with lower correlations, we can observe that the model captures well the characteristics of the lesion output. For instance, the indoor scene for $e_0$ obtains $r = 0.81$, however, it can be appreciated that the linear model accounts well for the disappearance of red pixels in the lesion output. This is also evident in the kite picture of $e_2$ where blue pixels have vanished or the bench picture of $e_5$ with green pixels.

Naturally, there are limits to this linear modelling. For instance, the excess of chromaticity (pink and blue colours) in the indoor scene of $e_6$ is not fully captured by its linear model. The most extreme can be observed in the kite picture of $e_3$ for the *rgb2lab* model (Figure D.3) where the non-linear nature of lesion output is not accounted for in the linear model. Nevertheless, these parsimonious

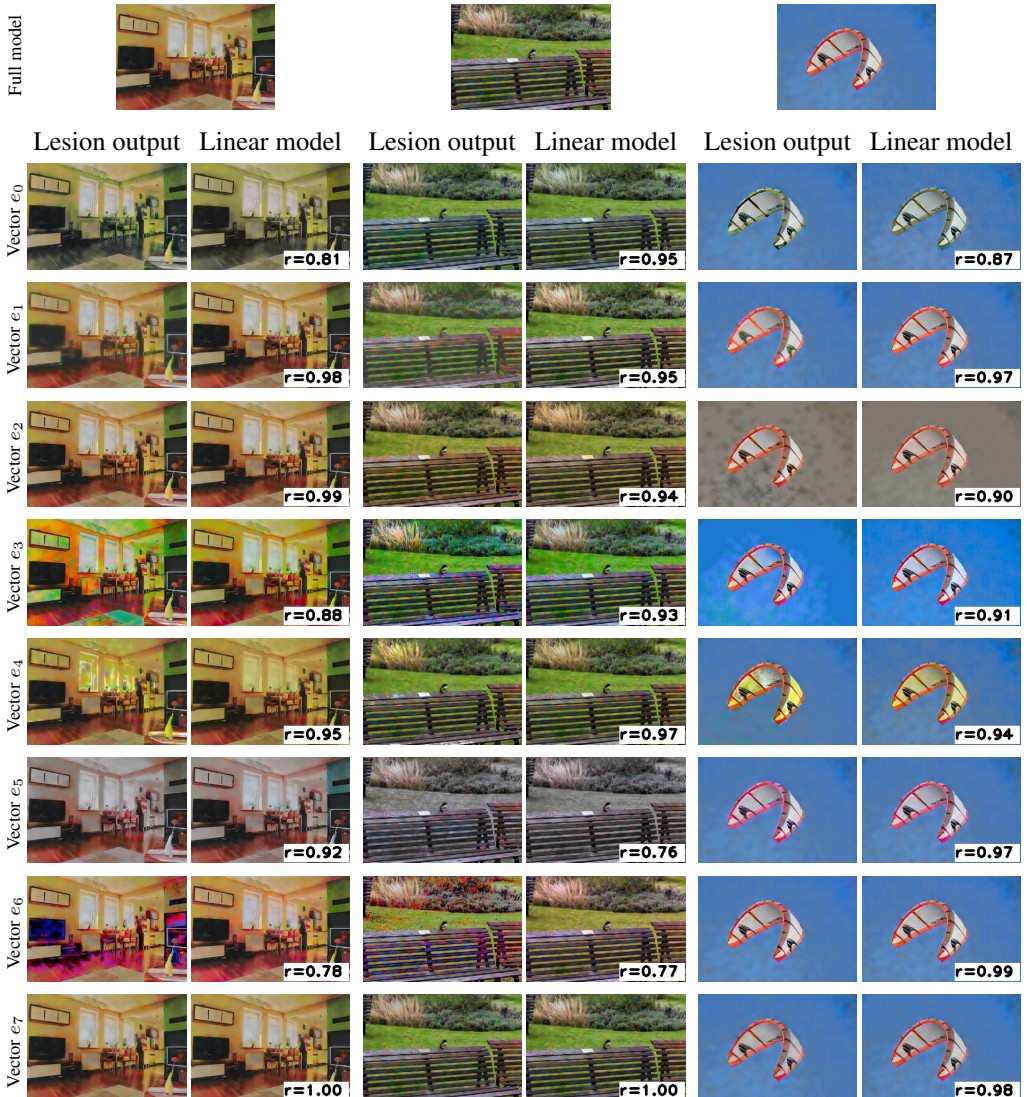

Figure D.2: The linear modelling of vector lesion for *rgb2dkl* VQ-VAE of **K=8** and **D=128**. The denoted $r$ on the bottom right corner of an image is the measure of transformations' fitness.

transformations reveal great details about the information encoded by each vector deserving more thorough investigation in future studies.

In Figure D.5 we have illustrated the impact of each linear transformation applied to the entire RGB cube. This gives an intuitive idea of what each vector performs in a simple glance. Absence of some vectors results in the collapse of a chromatic direction. Others shear, shrink or expand the colour space.

### D.3    HUE SHIFT

We quantified the chromatic shifts (in CIE L*a*b*) between the reconstructed image of full embedding space and lesion one. The differences computed for all pixels over a hundred random images from the COCO dataset are illustrated in Figure D.6.

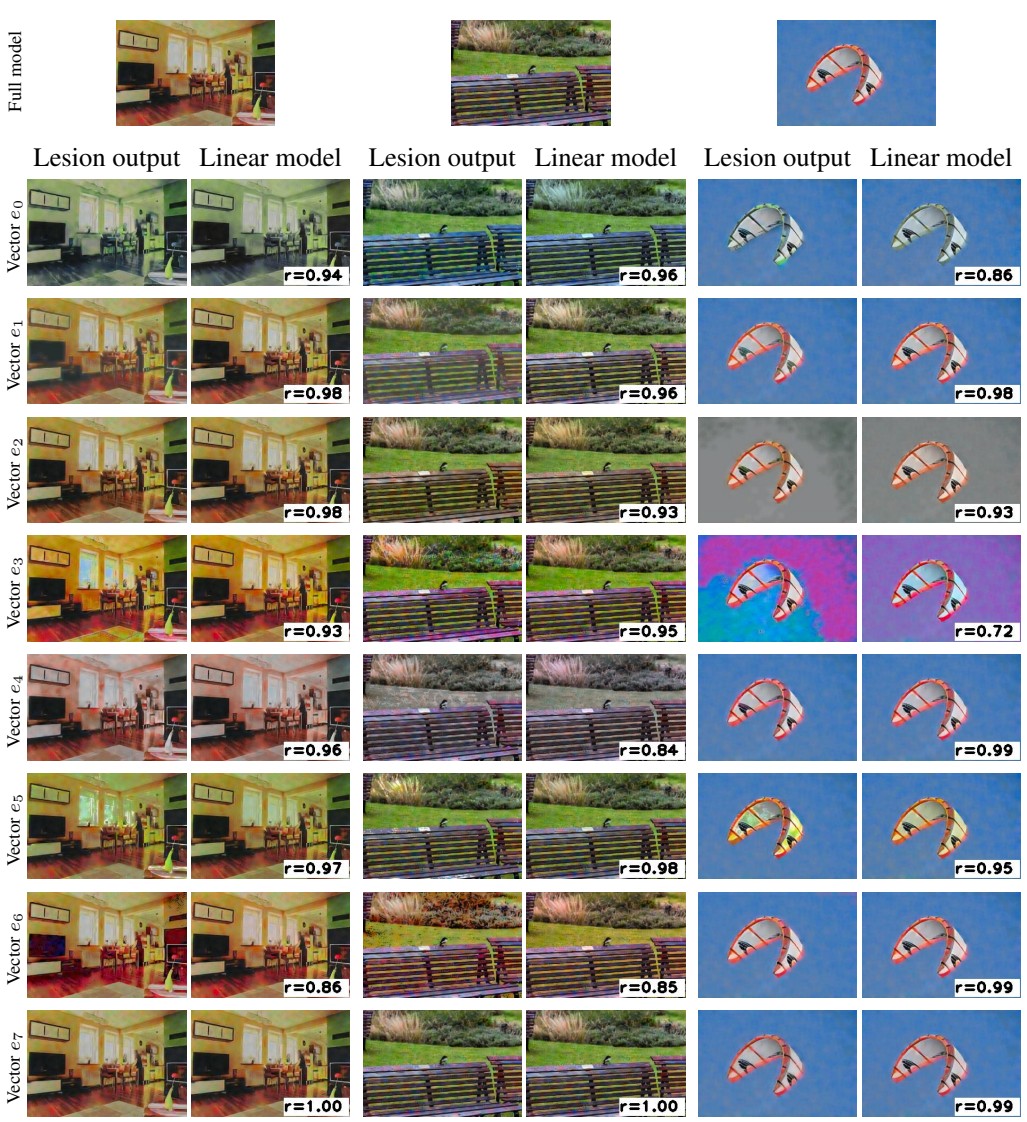

Figure D.3: The linear modelling of vector lesion for *rgb2lab* VQ-VAE of **K=8** and **D=128**. The denoted $r$ on the bottom right corner of an image is the measure of transformations' fitness.

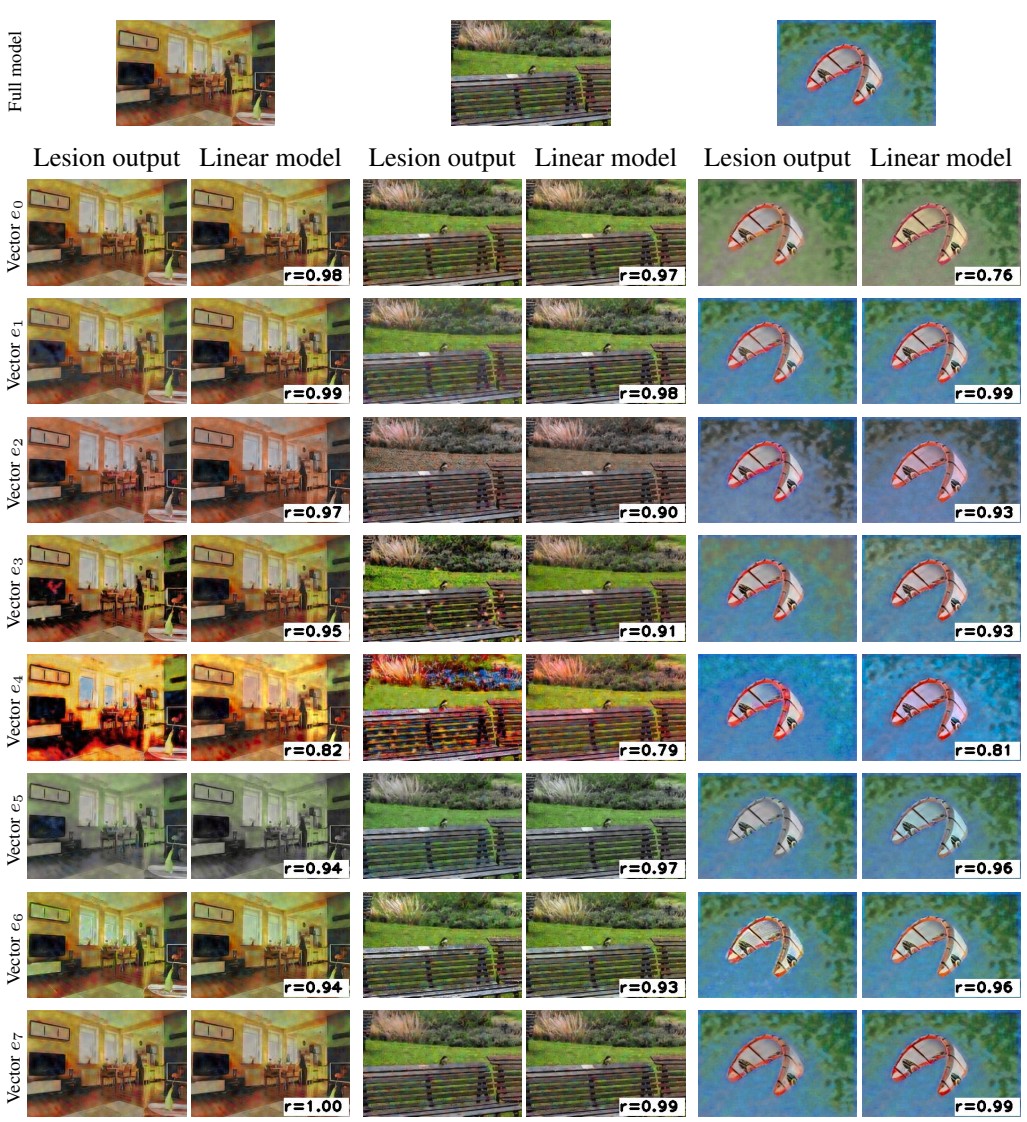

Figure D.4: The linear modelling of vector lesion for *rgb2rgb* VQ-VAE of **K=8** and **D=128**. The denoted $r$ on the bottom right corner of an image is the measure of transformations' fitness.

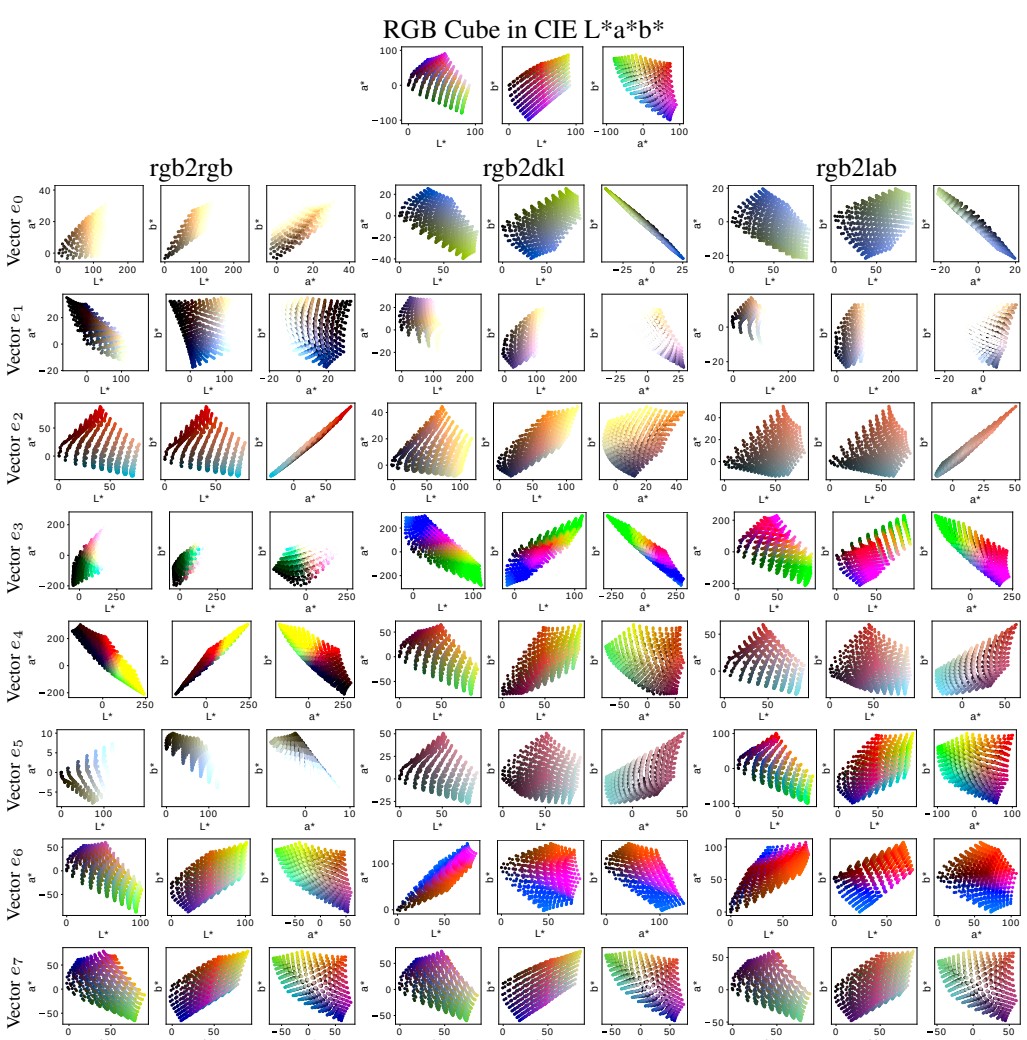

Figure D.5: The impact of vector lesion on RGB cube plotted in CIE L*a*b* coordinate,

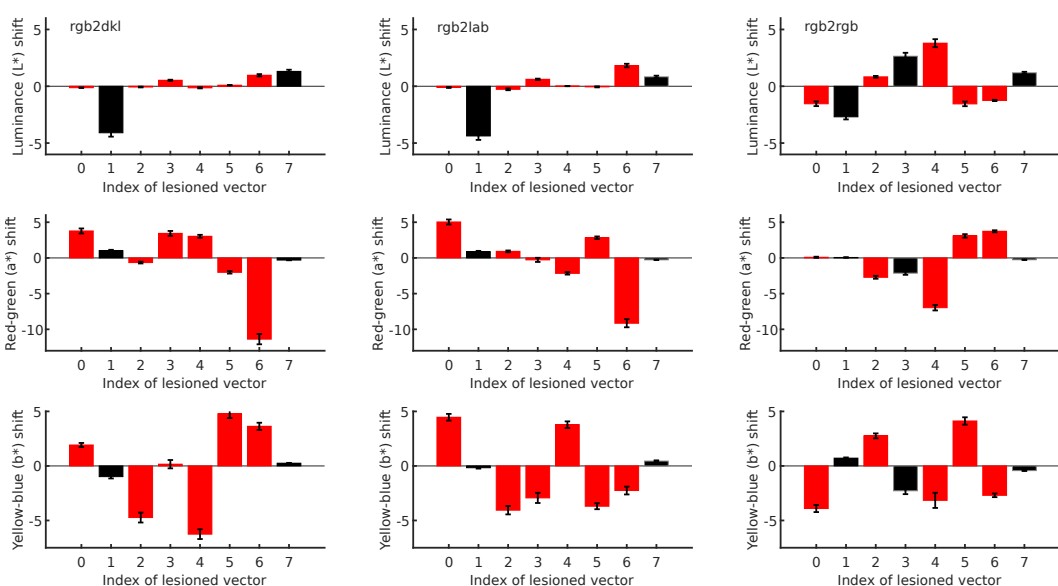

Figure D.6: Average hue shifts in CIE L*a*b* colour space for each vector lesion. Numbers in the $x$-axis denote the indices of the embedding vectors. All networks are of $\{e\} \in \mathbb{R}^{8 \times 128}$. Black- and red-bars indicate significant impact on the luminance- or chromatic-channels respectively.

