# OpenReview forum: "Learning Representation in Colour Conversion"
_ICLR.cc/2021/Conference — Reject_

### Official Review · AnonReviewer1 · 2020-10-28
**The motivation is unclear and non  additional knowledge is given**

**Rating:** 4
**Confidence:** 5

**Review:**

The motivation for this paper is quite hard to understand. A VQ-VAE is directly applied to convert an image from one colour space to another one. However, the colour space transform is human-defined, usually involving linear and a few non-linear (like selecting the maximum value is HSV) procedures. In this case, the latent space of VQ-VAE should be collapsed into this simple equation easily. The analysis of this paper does not teach us any additional knowledge.
The motivation of finding a better embedding space of colour is admirable, unfortunately, the analysis and methodology does not support the motivation.

---

> ### Author Response · Authors · 2020-11-16
> **Improving the motivation and highlighting take home message**
>
> Thank you for reviewing our article. We appreciate your constructive comments.
>
> ### Motivation
> This was also pointed out to us by the first reviewer.
> Thus, we have revised the manuscript to make our motivation clear. A framework for a fair comparison of different colour spaces within deep autoencoders that face a bottleneck to transmit information.
>
> > The motivation of finding a better embedding space of colour is admirable, unfortunately, the analysis and methodology does not support the motivation.
>
> In order to answer this research question, we aimed to design experiments that make the comparison across input-output colour spaces as fair as possible. Essentially, all factors are identical in the presented networks except the input-output colour spaces. We would highly appreciate it if you could please expand more on why you think the methodology and analysis fall short to study the stated motivation.
>
> ### What does the article teach us?
> We believe the findings of this article show (i) the choice of colour spaces can make a substantial impact on the encoding capacity of a VAE (i.e. networks with a bottleneck), thus from a practical perspective, applications that use the status-quo RGB perhaps could benefit from using opponent colour spaces. Furthermore as pointed out by the first reviewer the proposed framework is able to encompass additional constraints relevant in understanding why the considered representations could have emerged in the brain. (ii) The analysis of embedding space (with three different techniques) reach the same conclusion that each embedding vector encodes a certain type of information (a specific colour or luminance). We find this very exciting that within a complex deep network human-interpretable features emerge. (iii) A VAE consists of three stages (encoder, embedding space and decoder). To the best of our knowledge,  the contribution of each component to the network's task remains as an open question. Our results provide some evidence in the context of colour space conversion, the transformation occurs within the encoding stage. This perhaps triggers some ideas for other researchers conducted on VAEs with a different set of tasks.
>
> > The analysis of this paper does not teach us any additional knowledge.
>
> We appreciate it if you would please let us know whether you find our findings inadequate or you believe they are not grounded.
>
> Please see the revised manuscript where we have implemented these points (the largest changes are highlighted with blue colour). We would be grateful to receive more comments from you. Thanks a lot.

---

### Official Review · AnonReviewer4 · 2020-10-28
**review 3765**

**Rating:** 6
**Confidence:** 3

**Review:**

This paper proposes to study an interesting problem of how color informaiton is structured in the variational autoencoders (VAEs). Several instances of VAEs are trained in an unsupervised manner to perform color space conversion. Both low-level and high-level evaluations are performed to study the local statistics and global content of converted images. Several interesting conclusions are drawn from the experiments that help interpret the encoding process of autoencoders.

Overall, this paper studies an interesting problem and presents several insightful conclusions. My only concern is how significant the proposed method/task is, and how significant insights these conclusions could provide. To address this, it may show some applications based on the proposed task/conclusions.


I only have some minor issues.

1. It could be better to include results of state of the art methods on, e.g., object classification and scene segmentation. This could further show the potential application of proposed method.

2. encode -> encodes, Line 7, Page 2

3. Figure 1 is not referred to in the main text. Besides, it could be better to prodive more details in the caption.

---

> ### Author Response · Authors · 2020-11-16
> **Applied ColourConvnets**
>
> Thank you for reviewing our article. We appreciate your constructive comments.
>
> > To address this, it may show some applications based on the proposed task/conclusions.
>
> The potential applications for our framework could be any of those with standard VAEs. Essentially, by employing the proposed framework we showed that decorrelated colour spaces improve the encoding capacity of standard VAEs. The presented evaluation on high-level visual tasks could be considered as one example of such applications (i.e. an image-classification/scene-recognition embedding system with limited computational resources, thus requiring to perform under compressed images).
>
> A standard application for VAEs is image deblurring (e.g. this situation can arise with camera in motion). We simulated such a scenario by evaluating the ResNet50 accuracy with input images that are blurred with a Gaussian function. Similarly, we first passed the blurred images to our autoencoder (without any fine-tuning) and gave the output to ResNet 50. Here are the corresponding results:
>
> |  | $\sigma = 1.5$ | $\sigma = 2.5$ | $\sigma = 3.5$ |
> |: ---- | :---: | :---: | :---: |
> | Original images |0.57 | 0.37 | 0.22 |
> | rgb2rgb (128x128) |0.64 | 0.53 | 0.40 |
> | rgb2lab (128x128) |0.64 | 0.53 | 0.40 |
> | rgb2rgb (8x128) |0.44 | 0.25 | 0.11 |
> | rgb2lab (8x128) |0.54 | 0.40 | 0.24 |
>
> We can observe that when images are blurred, an object classification network benefit from ColourConvnets. Although for embedding space of 128x128 rgb2rgb is as good as the rgb2lab, for a smaller embedding space (8x128) with a greater rate of compression, only the rgb2lab can outperform the original images.
> The reduction of information for the embedding space 8x128 is 384 in bits: $\frac{224 \times 224 \times 3 \times 8} {56 \times 56 \times 1} = 384 $, where (the number 224 corresponds to the input image size, the number 3 denotes the number of colour channels, the number 8 corresponds to uint8 input images; the number 56 is the embedding space spatial size, the number 1 refers to the number of bits for vectors of embedding space with K=8).
> Given our manuscript including the supplementary materials is 21 pages, we have not included the Gaussian blur results in the revised version. However, if you think this would be interesting to many readers, we can include a corresponding figure.
>
> Last but not least, we believe our framework could potentially lead to applications of image compression applications. There have been some works (e.g. "Lossy Image Compression with Compressive Autoencoders" ICLR 2017) showing deep autoencoders might perform better than standard JPEG compression. However, incorporating our findings with such frameworks is outside of the scope of this article and we believe it can be addressed in future research.
>
>
> ### Minor issues:
> 1. We used state-of-the-art image classification (ResNet50) and scene segmentation (FPN) for high-level evaluation. Essentially, when these networks are input with the output of rgb2rgb ColourConvNet, they correspond to the state-of-the-art results. Thus, our baseline is always rgb2rgb of the same architecture. Nevertheless, ColourConvNets with embedding space of K=128 and D=128 obtain 70% top-1 accuracy on ImageNet and 60% IoU on COCO, on a par with the state-of-the-art on original images (it is worth considering that the reconstructed images are substantially compressed).
> To be precise, the pretrained ResNet50 we used obtains 73% accuracy on original images of ImageNet. The same network obtains 71% accuracy with the rgb2lab compressed images. The performance drop is only 2% for a reduction of about 55 in bits $\frac{224 \times 224 \times 3 \times 8} {56 \times 56 \times 7} \approx 54.85 $.
> 2. Thank you. We have corrected the typo.
> 3. Thank you for pointing this out. In the original submission, we only referred to the appendix figure. This has been corrected. We also expanded the corresponding caption with more details.
>
> Please see the revised manuscript where we have implemented these points (the largest changes are highlighted with blue colour). We would be grateful to receive more comments from you. Thanks a lot.

---

### Official Review · AnonReviewer2 · 2020-11-01
**Interesting assesment of color representations using autoencoders**

**Rating:** 7
**Confidence:** 5

**Review:**

Summary
-------------

The authors analyze the quality of known color representations (non-opponent and opponent) in terms of the quality of the reconstructed images when spatio-chromatic information is constrained by a bottleneck of a discrete (quantized) representation of reduced dimensionality.

The quantized representation and encoding-decoding transforms are defined by a loss function optimized through an autoencoding tool (in particular a Vector-Quantized Variational Autoencoder). The quality of the reconstructed images is measured in terms of (1) low-level similarity metrics (some of them with perceptual meaning), and (2) performance in higher-level tasks such as classification and segmentation.

The conclusion is that perceptually meaningful opponent representations such as DKL and CIELab are better suited to the imposed bottleneck as opposed to color representations related to retinal sensors such as RGB or LMS.

General opinion and recommendation
-----------------------------------------------------

I think the metodology and findings are really interesting to understand why the brain may have developed the opponent
representations that have better performance in the presented experiments. The use of an autoencoding tool to enforce the minimization of the loss suggests that comparison between the color representations is fair.

However, this message (pointing out the advantages of opponent representations wrt more trivial non-opponent representations) is not clearly stated. The biological meaningfulness of the selected constraints is not discussed either. Presentation is confusing at many points (see specific list below), but this can be fixed. Therefore,  I think the work should be accepted after proper clarifications and removing some misconceptions.

Major Points
----------------

The goal of the paper (in my view, a fair comparison of different color spaces in a bottleneck context using an appropriate optimization tool) is not clearly stated.

Instead some sentences in the abtract and introduction of the paper suggests that color representation is learnt. For instance, in the abstract and intro it is said "We propose a novel unsupervised task —colour conversion— to explicitly examine the colour representation learnt by deep networks (referred to as ColourConvNets)." and "the structure of internal representation provides insights on how this [color] transformation is performed within a neural network". At this point the reader may think that the proposed autoencoder will learn a specific color representation well suited for certain goal(s). However, the autoencoder is not learning color representations (they are imposed at input and output), the autoencoder is only imposing certain bottleneck and hence, it is a controlled way to assess the suitability of the considered color representations in constrained settings. In fact, spatial and chromatic parts of visual information are mixed in the vectors of the inner representation of the considered autoencoders. This (hard to interpret) mixture necessarily comes from the reduced dimension of the vectors. Therefore, strictly speaking, there is no "pure color representation learnt" in the inner representation of the autoencoder. This misunderstanding about "learning a color representation" when it is actually a spatio-chromatic representation imposed by the constraints and the selected input-output color spaces happened to me, and only disappeared at the end of page 3 and 4. This misunderstanding should be avoided in abstract and intro.

Another confusing description is talking about the "correlation" and "decorrelation" properties when presenting the considered spaces (in section 2.2) just after talking about learning efficient representations through the autoencoder. Mentions to the "efficiency" of color space through citations to [Buchsbaum83,Ruderman98,Lee01] should appear later in the discussion (not as early as in section 1.1).
Readers aware of Barlow's efficient coding hypothesis that leads to transforms that favour decorrelation and equalization
(which in color lead to PCA-like transforms [Buchsbaum83], and nonlinear equalizations that explain chromatic adaptation [Laparra12]) may wonder why the cost function did not include decorrelation or independence measures. I would suggest talking about the decorrelation properties only in the discusion (do not mention in section 1.1 and remove the left part of fig 2 -devoted to highlight correlation and decorrelation-, and make these points in an expanded "performance advantage" section). Actually, decorrelation and equalization properties of these color spaces has been measured in information theoretic units by Foster et al. 2008 and by Malo 2020. I think these references [Buchsbaum83,Ruderman98,Lee01,Foster08,Laparra12,Malo20] should be included in this decorrelation-equalization discussion.

Another confusing statement is the apparent contradiction between this statement "the conversion of RGB images
into other colour spaces yield to no performance improvement in ImageNet (Mishkin et al., 2017)" and the interesting
findings done in this work. It is important to stress that maybe results in (Mishkin et al., 2017) were not subject to big enough dimensionality constraints and then proper representation of color was not that relevant.

It is unclear how Fig 8 and Fig. C.1 were computed. To me this is really important to clarify the non-trivial mixture of spatial and chromatic information in the vectors. From the text, I guess, a single codevector was used to decode the image.
But, given the dimension of the codevectors (bigger than 3), they encode not only chromatic information but also spatial information. Then, how is it possible to obtain uniform color (as in figs 8 and C.1) with no spatial variation from a single codevector?

Minor Points
------------

* it is important to stress that the advantage of the proposed metodology to compare color spaces is that additional constraints can be included (such as entropy, energy, wiring, etc...). This framework able to encompass additional constraints is relevant to understand why the considered representations could have emerged in the brain.

* Correlation = 1 between L and M seems like too much. Is this correct?

* The value of the loss function is comparable in the different cases after training? This would be necessary for a fair comparison, isnt it?

References:
----------------

[Foster08] D.H. Foster, I. Marin-Franch, and S.M.C. Nascimento. 2008. Coding efficiency of CIE color spaces. In Proc. 16th Color Imag. Conf. Soc. Imag. Sci. Tech., 285–288

[Laparra12] V. Laparra, S. Jiménez, G. Camps and J. Malo. 2012. Nonlinearities and adaptation of color vision from sequential principal curves analysis. Neural Computation 24, 10 (2012), 2751–2788

[Malo20] J. Malo. Information Flow in Color Appearance Neural Networks. Accepted in Entropy Conference 2020 https://arxiv.org/abs/1912.12093

---

> ### Author Response · Authors · 2020-11-16
> **Constructive Feedback in Details**
>
> Thank you very much for reviewing our article. We highly appreciate your constructive and detailed comments. They were truly helpful in improving the quality of our manuscript.
>
> > However, this message (pointing out the advantages of opponent representations wrt more trivial non-opponent representations) is not clearly stated.
>
> In the revised version, we have emphasised the advantages of opponent colour spaces with respect to their encoding capacity. We also have tried to better highlight the other contribution. The comparison of the VQ-VAE's internal colour representation across colour spaces reveals what each embedding vector has encoded and provides some cues on where the colour transformation might occur (in the encoder).
>
> > The biological meaningfulness of the selected constraints is not discussed either.
>
> In the revised version, while explaining more on biological meaningfulness of the selected constraints, we tried to make sure it remains accessible to interested readers from other communities.
>
> ## Major points
>
> > The goal of the paper (in my view, a fair comparison of different color spaces in a bottleneck context using an appropriate optimization tool) is not clearly stated.
>
> Thank you for pointing out this to us. We have revised the abstract, introduction and Section 2 accordingly emphasising this point. We have stated clearly that the proposed framework allows for a fair comparison across colour spaces within a system faced with a constraint on its information flow.
>
> > Instead some sentences in the abtract and introduction of the paper suggests that color representation is learnt. [...] Therefore, strictly speaking, there is no "pure color representation learnt" in the inner representation of the autoencoder. [...]
>
> We appreciate you raising this point.
> The input-output colour spaces are imposed on the network (in the revised version, we have mentioned this upfront in the second paragraph of the introduction to avoid any confusions). ColourConvNets learn to efficiently compress visual information and to perform the colour conversion. We completely agree that there is no “pure colour representation learnt”, and as you pointed out the learnt representation is spatio-chromatic. However, the results presented in section “Interpreting the Embedding Space” show embedding vectors are associated with certain colours and their removal results in the disappearance of certain colours. We quantified this with the linear modelling (Section 5.2), the error of this modelling suggests a large part of encoded information in each embedding vector is related to colour features independent of the spatial component (the linear model is applied to all pixels equally). But of course, you are completely right and the complete representation is spatio-chromatic (please find our response regarding Fig 8). We have attempted to clarify this in the revised version.
>
> > Another confusing description is talking about the "correlation" and "decorrelation" properties [...]
>
> Thank you for your suggestion. In the revised version, we have moved the correlation-decorrelation terminology to the expanded "performance advantage" section. We found the suggested references very relevant and have incorporated them. Figure 2 was also revised accordingly.
>
> >  [...] Mishkin et al., 2017
>
> This is an important point, thank you for pointing this out. Mishkin et al. (2017) analysed AlexNet like architecture without any bottlenecks. We have discussed this in the revised performance advantage section.
>
> > Fig 8 and Fig. C.1
>
> Fig 8 was created by sampling from the embedding space an example of spatial size 2x2 with all cells set to the same vector index (e.g. [[0,0], [0,0]] corresponding to vector 0). The resulting reconstructed image is an 8x8 image. Previously we had averaged over these pixels for visualisation purposes, thus they were 100% uniform. In the revised version, we have removed this averaging to avoid any confusion (although many of them still appear very uniform since the spatial variation is tiny).
> This is explained in Section 5.1 and a new figure for a different combination of vectors in the horizontal, diagonal and vertical direction has been added.
>
> ### Minor points
>
> 1. We added this statement in Section 2 and are looking forward to incorporating these constraints in our framework for future research (thanks a lot for the tip).
> 2. The correlation between L and M is 0.9997. In the manuscript we have rounded them in the second decimal, thus it has become 1.00. We have changed the equal sign to approximately equal to avoid any confusions.
> 3. We have added a figure (B.1) regarding the loss functions which shows convergence across networks are comparable.
>
> Please see the revised manuscript where we have implemented these points (the largest changes are highlighted with blue colour). We would be grateful to receive more comments from you. Thanks a lot.

---

### Decision · Program_Chairs · 2021-01-07
**Final Decision**

**Decision:**

Reject

**Comment:**

This paper proposes a novel unsupervised task of colour conversion. In this respect, the task
becomes more like a regression problem -- rather than autoencoding the decoder needs to reconstruct the pixels in a different color system.
While the idea is potentially interesting, there are fundamental problems with the paper:

* The motivations of the paper are obscure, (understanding colour representation in complex visual systems? Learning better representations? Disentangling color related information from the rest)
* No analysis is provided to highlight what the novel objective is achieving

The answers of the authors to AnonReviewer1 are not very convincing. As AnonReviewer1 has pointed out, the mapping between the color spaces is typically a simple invertible map so any conclusion that the authors arrive about ‘substantial impact’ could be simply the artefact of the particular architecture choice. The other claim, that ‘the proposed framework is able to encompass additional constraints relevant in understanding why the considered representations could have emerged in the brain’ quite far fetched and speculative at best.

The authors have a point in their reply ii) to AnonReviewer1 but if the claim is about the particular color coding schemata, it would be natural to include simple experiments where some arbitrary 3x3 invertible mapping (e.g. rgb in spherical or cylindrical coordinates) next to other color schemata to make a stronger point.

In point iii) the authors refer to tasks without being very explicit about what the tasks are. Colorization is a known proxy pretext task for learning representations when the downstream classification task is not known a-priori. The paper would have been much more easy to motivate if the authors could demonstrate the merit of the proposed objective using a more extensive and careful representation learning evaluation methodology.

In light of the above points, I feel that the paper needs further iterations to be presented at ICLR.